# Rethinking and Improving Multi-task Learning for End-to-end Speech Translation

**Yuhao Zhang[1], Chen Xu[1], Bei Li[1], Hao Chen[1],**
**Tong Xiao[1,2]\*, Chunliang Zhang[1,2] and Jingbo Zhu[1,2]**
[1]School of Computer Science and Engineering,
Northeastern University, Shenyang, China
[2]NiuTrans Research, Shenyang, China
yoohao.zhang@gmail.com, {xuchenneu,libei_neu}@outlook.com
{xiaotong, zhangcl, zhujingbo}@mail.neu.edu.cn

## Abstract

Significant improvements in end-to-end speech translation (ST) have been achieved through the application of multi-task learning. However, the extent to which auxiliary tasks are highly consistent with the ST task, and how much this approach truly helps, have not been thoroughly studied. In this paper, we investigate the consistency between different tasks, considering different times and modules. We find that the textual encoder primarily facilitates cross-modal conversion, but the presence of noise in speech impedes the consistency between text and speech representations. Furthermore, we propose an improved multi-task learning (IMTL) approach for the ST task, which bridges the modal gap by mitigating the difference in length and representation. We conduct experiments on the MuST-C dataset. The results demonstrate that our method attains state-of-the-art results. Moreover, when additional data is used, we achieve the new SOTA result on MuST-C English to Spanish task with 20.8% of the training time required by the current SOTA method.

## 1 Introduction

End-to-end (E2E) models have made significant strides in the artificial intelligence realm, especially in speech translation (ST). These models have low latency and less error propagation by providing direct translations from speech inputs (Bérard et al., 2016; Duong et al., 2016). This approach contrasts with traditional pipeline models that rely on separate automatic speech recognition (ASR) and machine translation (MT) systems. However, the E2E model's single-model design poses new challenges due to its need for cross-modal and cross-lingual transfers (Zheng et al., 2021; Xu et al., 2021). To address this, recent studies have utilized multi-task learning (MTL), leveraging cross-modal or cross-lingual training objectives for pre-training or joint

---
*Corresponding author.

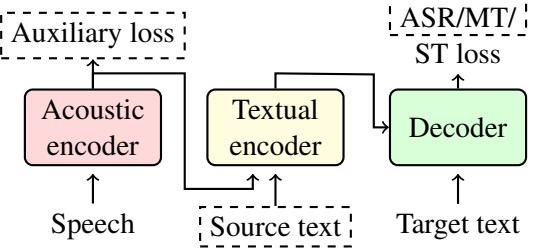

Figure 1: Multi-task training architecture for Speech translation. The dashed line part will be removed in fine-tune stage.

training (Tang et al., 2021; Ye et al., 2021; Dong et al., 2021b; Han et al., 2021). This technique assures good convergence of the models and emphasizes the importance of the auxiliary loss, offering a brand-new perspective for further advancements in E2E ST (Tang et al., 2022).

However, further exploration is necessary to determine how and to what extent these auxiliary tasks aid the final ST model. Notably, not all auxiliary tasks in the fine-tuning stage are beneficial. This inconsistency arises because MTL is typically viewed as a multi-objective optimization problem, often resulting in training trade-offs when objectives conflict (Désidéri, 2012). The ideal MTL outcome is to achieve Pareto optimality (Sener and Koltun, 2018), indicating solutions are superior to any alternatives. Since MTL does not ensure optimal performance for the ST task, fine-tuning is crucial to overcome this shortcoming. Some studies even underscore a critical conflict between the ST task and ASR and MT tasks (Xu et al., 2021; Tang et al., 2022), necessitating a fine-tuning strategy. So answering these questions is crucial to designing an optimal MTL strategy for the ST task.

In this paper, we rethink task consistency in MTL and introduce a gradient-based consistency metric, which denotes the consistency of the gradient direction between the ST task and other auxiliary tasks. Our analysis shows that 1) ASR aids the acoustic

encoder and MT facilitates the textual encoder in audio-to-text transfer, 2) length inconsistency hinders aligning the representations of the two modalities, 3) disparity between noisy speech features and clean text embeddings as considerable obstacles, and 4) the timing and degree of task influence exhibit significant variation.

Inspired by the aforementioned observations, we relax the ASR task that only uses it to help the acoustic encoder. We propose the Looking-Back Mechanism (LBM) to overcome length consistency. It can significantly shrink the speech length without information loss. To bridge the modality gap, we introduce the Local-to-Global (L2G) training strategy. By incorporating speech-like noise into the text and utilizing an L2G extractor, we enhance contextual information at each layer. This method effectively guides the interaction between audio and text sequences and aligns with audio processing requirements. We further propose a task-norm-based weight decrease method to speed up training, adjusting task weights based on auxiliary task influence and timing, thus avoiding unnecessary training costs.

We test our method on MuST-C 8 language datasets. The results show our method can achieve comparable performance with SOTA work without fine-tuning on ST task. We then set the new SOTA by utilizing the knowledge distillation method (Liu et al., 2019). With additional data, we achieve comparable performance with that of using only $12.5\% \sim 33.3\%$ training cost compared with the SOTA work [1].

## 2 Task Consistency Quantification

In this section we investigate the consistency issue of multi-task learning on three tasks. We randomly sample $n$ samples as analysis set $\mathcal{D} = \{(\mathbf{s}, \mathbf{x}, \mathbf{y})\}$. Here $\mathbf{s}, \mathbf{x}, \mathbf{y}$ denote the speech, transcription and translation sequences respectively. We then build the ASR set $\mathcal{D}_{\text{ASR}} = \{(\mathbf{s}, \mathbf{x})\}$ and MT set $\mathcal{D}_{\text{MT}} = \{(\mathbf{x}, \mathbf{y})\}$. By inputting the same data $(\mathbf{s}, \mathbf{x}, \mathbf{y})$ into the model, we observe that different training tasks yield distinct parameter gradients. The losses of different tasks are given by:

$$\mathcal{L}_{\text{ST}} = -\sum_i^{|\mathbf{y}|} \log p(y_i|y_{1:i-1}, \mathbf{s}) \qquad (1)$$

$$\mathcal{L}_{\text{ASR}} = -\sum_i^{|\mathbf{x}|} \log p(x_i|x_{1:i-1}, \mathbf{s}) \qquad (2)$$

$$\mathcal{L}_{\text{MT}} = -\sum_i^{|\mathbf{y}|} \log p(y_i|y_{1:i-1}, \mathbf{x}) \qquad (3)$$

where $|\cdot|$ denotes length of the sequence. The model contains three modules: the acoustic encoder (A-Enc), the textual encoder (T-Enc), and the decoder. The MT task shares the T-Enc and the decoder with the ST while the ASR shares all parameters. Thus we can quantify the consistency of different tasks from the perspective of the gradient.

We employ cosine similarity as the metric for gradient direction, where higher values indicate stronger consistency between tasks within a single model. To calculate the similarity, we flatten the gradient matrix into a vector, providing a more accurate assessment of task consistency, despite yielding lower values. We focus on evaluating the gradient of the feed-forward (FFN) sub-layer and self-attention (ATTEN) sub-layer as representative parameters of the entire model. Our backbone model is the robust ConST (Ye et al., 2022). In our experiments, we set $n = 200$. We sample five times and use average values to obtain solid results.

### 2.1 Consistency in Different Modules

The consistency between the ST task and the other two tasks (ASR and MT) within these modules is shown in Figure 2. Although the ASR task shares all the parameters with the ST task, only the A-Enc exhibits high consistency with the ST task. This indicates that modeling speech in the A-Enc serves the same purpose for both ASR and ST tasks, which aligns with the conclusions of Anastasopoulos and Chiang (2018) and Bahar et al. (2019). However, the consistency between the two tasks decreases sharply after the textual modal processing in the T-Enc and decoder. The decoder's divergence is expected due to generating texts in different languages. The T-Enc converts the acoustic feature to a textual feature for both tasks, but Figure 2 reveals lower consistency. It suggests a specific need for semantic-level representation in the ST task to achieve the cross-lingual goal.

The decoder exhibits higher consistency compared to the encoder for the MT task, suggesting that the behavior of the ST decoder leans towards cross-language processing. However, the T-Enc still exhibits low consistency. Taking into account the above analysis on the ASR task, the T-Enc plays

[1]Our code is available at https://github.com/xiaozhang521/IMTL.

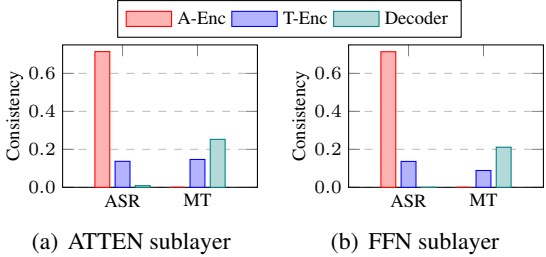

(a) ATTEN sublayer      (b) FFN sublayer

Figure 2: Consistency of different tasks in different modules.

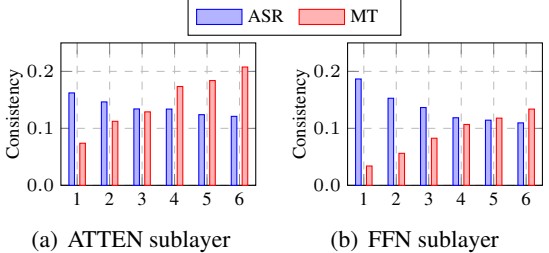

(a) ATTEN sublayer      (b) FFN sublayer

Figure 3: Consistency of different tasks in different layer of T-Enc.

a unique role in MTL and does not align closely with either of the other two tasks (Xu et al., 2021). Therefore, our subsequent analysis will focus on the T-Enc.

## 2.2 Impact of T-Enc

We conducted a comparison of the consistency between the ST task and the other two tasks within each layer of the T-Enc. Figure 3 illustrates that the bottom layer of the T-Enc demonstrates a stronger resemblance to the ASR task. The discrepancy arises as the feature extraction process diverges further between the ASR and ST tasks. This suggests that the cross-modal transformation of A-Enc is not achieved, then the T-Enc begins extracting textual information which is required by ST to adequately address the cross-lingual objective. We also noticed that the ST task gradually aligns with the MT task, but the consistency between them is still low. This raises the question of whether the ASR task leads to insufficient alignment between speech and text in the T-Enc.

We conducted further investigations into the impact of the ASR task on the T-Enc. We introduced two widely used ASR losses: 1) the Connectionist Temporal Classification (CTC) loss (Graves et al., 2006) after the A-Enc, and 2) the cross-entropy (CE) loss after the decoder. The former updates only the A-Enc (Bahar et al., 2019), while the latter updates the entire model. By exploring various MTL combinations, the changes in consistency between the MT and ST tasks are depicted in Figure 4. We discovered that as the ASR training becomes more intensive, the decrease in consistency between the MT and ST tasks becomes more pronounced at the top layers. Although increasing the ASR training workload burdens the T-Enc, research indicates that the ASR task is crucial in helping the acoustic encoder model speech (Le et al., 2023). We find the impact of the ASR task on the T-Enc is limited. Thus we further investigate other factors that impede the consistency between the MT and ST tasks.

## 2.3 Discrepancy between MT and ST

We focus on two main differences between speech and text features: length and representation space. The length disparity arises from modeling granularity (frames for speech while sub-words for text) (Xu et al., 2023b). The representation space discrepancy is due to acoustic features extracted by the acoustic encoder lacking text-based information (Li et al., 2021; Fang et al., 2022). We implement the shrinking method (Liu et al., 2020; Dong et al., 2021a) and contrastive learning (CL) loss (Ye et al., 2022; Zhang et al., 2023) at the top of T-Enc respectively to investigate the two issues. we employ "Length" and "Rep" to represent the shrinking and CL methods respectively in Figure 5 and 6.

To remove the influence of the ASR task, the experiments are conducted with the MT and ST tasks. Figure 5 illustrates the shrinking method effectively increases consistency in the decoder, as a more compact sequence is easier to extract information from during cross-attention. However, this approach also results in the loss of original information, leading to a significant degradation in the consistency between the two tasks in the T-Enc. On the other hand, when incorporating additional alignment loss, the changes in consistency within both modules are minimal.

To explore why CL loss does not work, we conduct an in-depth analysis from the perspective of information entropy (IE). Higher entropy implies greater outcome uncertainty. We compute the IE of each T-Enc's self-attention weights, as shown in Figure 6. The MT task shows lower IE compared to the ST task, indicating reduced noise sources in its representation. When we shrink the length of the speech, the IE is noticeably reduced. This can

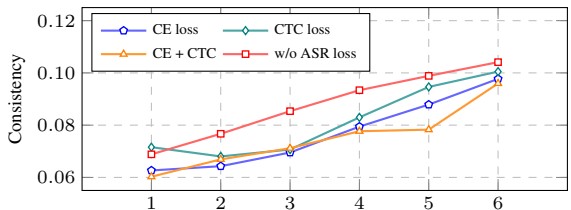

Figure 4: Influence of textual encoder by ASR training strategy.

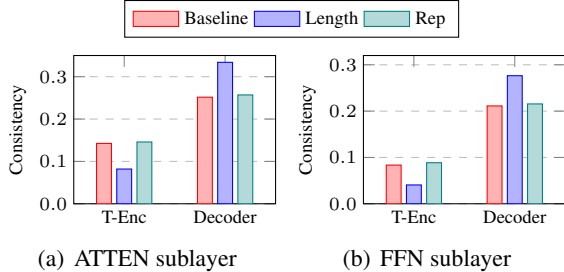

Figure 5: Consistency of different modules.

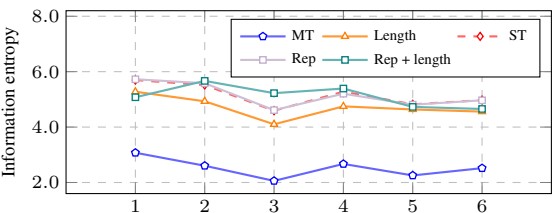

Figure 6: Information entropy of T-encoder attention weight.

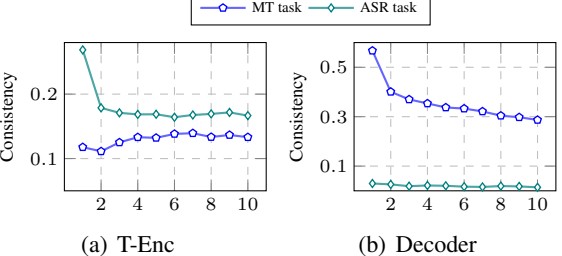

Figure 7: Changes of consistency along training epochs.

explain why the decoder exhibits higher gradient consistency. But if we use the CL loss, it does not have a impact on the IE. The noisy speech sequence makes it difficult to learn more textual information. This finding can explain the CL on sentence-level representations performs worse than token-level approaches (Ye et al., 2022). However, when the CL loss is introduced based on the compressed speech, we observe that additional information is learned in the middle layers. Thus we can find that shrinking is necessary and information gap between ST and MT tasks still needs to be mitigated.

## 2.4 Time of Taking Effect

We have identified the discrepancies between different tasks, but the interplay of these tasks during the training process has not been thoroughly studied. Figure 7 illustrates the changes in consistency among the different modules throughout training. We observe that the assistance provided by the ASR task primarily occurs at the early stage and becomes less significant later on. On the other hand, the impact of the MT task on the ST task is more complex, with a gradual decrease in consistency over time, indicating slower assistance to the ST task. Additionally, the behaviors of the T-Enc and decoder differ significantly. These observations highlight the diversity in the timing and effects of different tasks, underscoring the need for a careful strategy to optimize their training effects and timing.

## 3 Method

We propose the improved MTL (called IMTL) method from three perspectives: 1) denoising the ST sequence, 2) adding noising to MT, and 3) and improving training efficiency. The overall training objective is:

$$\mathcal{L} = \mathcal{L}_{ST} + w_a\mathcal{L}_{ASR} + w_m\mathcal{L}_{MT} + w_c\mathcal{L}_{CL} \quad (4)$$

The CL denotes contrastive learning and we set the $w_c$ to 0.3. Based on the above analysis, we use the CTC loss as the ASR task.

## 3.1 Stable Shrinking

From the previous analysis, we find the decoder benefits from the decrease in speech length. Though Liu et al. (2020) and Dong et al. (2021a) have proposed methods to figure the issue out, there are two main problems still need to be improved.

**Instability** Once tokens are removed, the gradient from the decoder can not guide the acoustic encoder. Especially when the prediction of speech is not accurate at the earlier training stage, this will cause the unstable training.

**Information loss** If tokens are wrongly removed by method, information loss will happen. The blank token also contains pause information which can help the model understand the sentence.

To address the aforementioned issues, we propose the Looking-back mechanism (LBM), as depicted in Figure 8. Given a sequence of speech

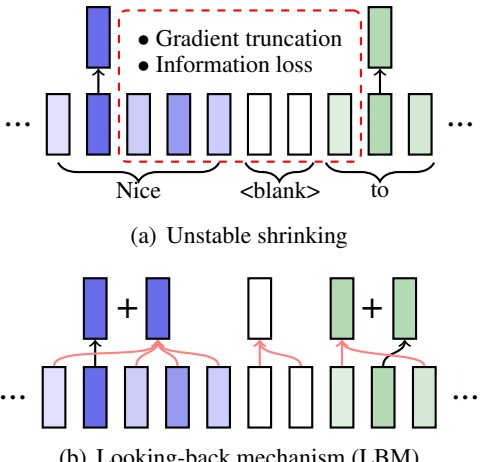

(a) Unstable shrinking

(b) Looking-back mechanism (LBM)

Figure 8: (a) Unstable shrinking and (b) LBM. The red line in (b) can supply additional information. Thus it avoids the two problems in (a).

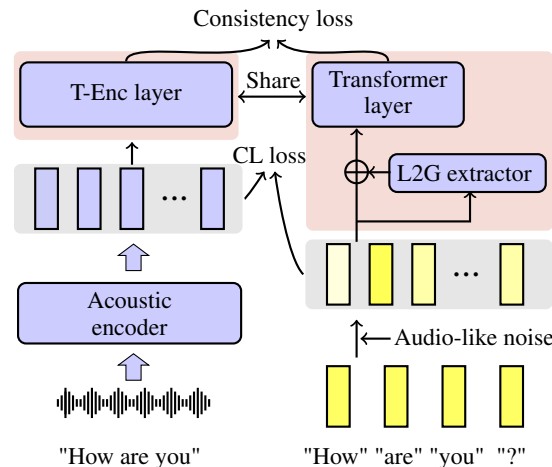

Figure 9: Encoder of Local-to-global (L2G) training. One T-Enc layer consists of Transformer layer and L2G extractor. The light pink parts are shared.

features $\mathbf{s} = (s_1, ..., s_n)$, we first calculate the probabilities of CTC paths and extract the position with the highest value as the decoding result $T = (t_1, ..., t_n)$. Since $T$ is generated through monotonic decoding, adjacent positions may contain repeated tokens in the result. For each repeated segment in the sequence, we select the token with the highest confidence within that segment to form a new unique result $T' = (t'_1, ..., t'_m)$. Additionally, the corresponding new features $\mathbf{s}' = (s'_1, ..., s'_m)$ can be generated. Note that we do not filter out the blank positions to prevent error propagation. Therefore, the key to resolving the mentioned problems lies in effectively transferring all the information from $\mathbf{s}$ to the compressed features $\mathbf{s}'$. The LBM method utilizes features from $\mathbf{s}'$ to retrospectively retrieve information from $\mathbf{s}$ and extract the missing information.

Formally, for an arbitrary feature $s'_i$ in $\mathbf{s}'$, we can determine its index $j$ in $\mathbf{s}$ based on $t'_i$ and $T$. We set a boundary $b$ for looking back, ensuring that $[j - b, j + b]$ contains all the repeated tokens. We construct the search matrix $A$ by including all features from $\max(1, j - b)$ to $\min(j + b, n)$, excluding the feature at index $j$. We then search and aggregate information in $A$ using the following formula:

$$\tilde{s}_i = \mathrm{Softmax}\left(\mathcal{R}(s'_i) \cdot \mathcal{R}(A)^{\mathrm{T}}\right) \cdot A \quad (5)$$

where the $\mathcal{R}$ denotes the linear transfer, $\mathrm{Softmax}()$ normalizes the correlation between $s'_i$ and $A$ to $0 \sim 1$. We final use the fusion module to integrate

the $s'_i$ and $\tilde{s}_i$:

$$s_i^f = \mathrm{FFN}\left(\mathrm{Norm}(s'_i + \tilde{s}_i)\right) \quad (6)$$

here $\mathrm{Norm}()$ denotes the normalization layer, and $\mathrm{FFN}()$ denotes a feed-forward network used to filter redundant information. The LBM method can automatically learn the weights of each repeated frame, ensuring that the obtained $s_i^f$ does not introduce additional noise. Even when the length is significantly reduced, the LBM can preserve all the original information and avoid gradient truncation, thereby promoting stable training.

## 3.2 Local-to-global Training

We have observed an information difference in representation between the MT and ST tasks in the T-Enc. The main reason is that the speech feature undergoes high-level abstraction by the acoustic encoder, while the text embedding remains unprocessed and devoid of any noise. This inherent difference causes the model to classify these two tasks differently, resulting in inconsistent gradients. Wang et al. (2020b) injects some audio-like tokens into the MT sequence, while we propose the local-to-global (L2G) training strategy to bridge the information gap.

We first introduce noise to the clean text embedding. Taking into account the characteristics of repeated information and blank positions in speech sequences, we randomly add blanks or duplicate certain tokens with a probability of 0.2 for each position. Our goal is to facilitate the learning of consistent representations for the two tasks. To

| Models | FT | En-De | En-Es | En-Fr | En-It | En-Nl | En-Pt | En-Ro | En-Ru | Avg. |
|---|---|---|---|---|---|---|---|---|---|---|
| Fairseq ST[†] (Wang et al., 2020a) | - | 22.7 | 27.2 | 32.9 | 22.7 | 27.3 | 28.1 | 21.9 | 15.3 | 24.8 |
| Revisit ST[†] (Zhang et al., 2022a) | ✓ | 23.0 | 28.0 | 33.5 | 23.5 | 27.1 | 28.2 | 23.0 | 15.6 | 25.2 |
| STEMM (Fang et al., 2022) | ✓ | 25.6 | 30.3 | 36.1 | 25.6 | 30.1 | 31.0 | 24.3 | 17.1 | 27.5 |
| ConST (Ye et al., 2022) | ✓ | 25.7 | 30.4 | 36.8 | 26.3 | 30.6 | 32.0 | 24.8 | 17.3 | 28.0 |
| M³ST (Cheng et al., 2022) | ✓ | 26.4 | 31.0 | 37.2 | 26.6 | 30.9 | 32.8 | 25.4 | 18.3 | 28.6 |
| CMOT (Zhou et al., 2023) | ✓ | 27.0 | 31.1 | 37.3 | 26.9 | 31.2 | 32.7 | 25.3 | 17.9 | 28.7 |
| CRESS (Fang and Feng, 2023) | ✓ | 27.2 | **31.9** | 37.8 | 27.3 | 31.6 | 33.0 | **25.9** | **18.7** | 29.2 |
| Baseline | ✓ | 25.8 | 30.4 | 36.7 | 26.1 | 30.5 | 32.0 | 24.7 | 17.3 | 28.0 |
| IMTL | - | 26.9 | 31.5 | 37.7 | 27.3 | 31.3 | 33.0 | 25.5 | 18.3 | 28.9 |
| IMTL-KD | - | **27.5** | 31.8 | **38.2** | **27.7** | **32.0** | **33.4** | **25.9** | 18.6 | **29.4** |

Table 1: Performance on different data set. FT denotes the model needs fine-tuning stage. † means the work does not use the unlabeled speech data.

| Models | FT | En-De | En-Fr | En-Es |
|---|---|---|---|---|
| ConST (Ye et al., 2022) | ✓ | 28.3 | 38.3 | 32.0 |
| STPT (Tang et al., 2022) | ✓ | - | 39.7 | 33.1 |
| M³ST (Cheng et al., 2022) | ✓ | 29.3 | 38.5 | 32.4 |
| CMOT (Zhou et al., 2023) | ✓ | 29.0 | 39.5 | 32.8 |
| CRESS (Fang and Feng, 2023) | ✓ | 29.4 | 40.1 | 33.2 |
| SpeechUT (Zhang et al., 2022b) | ✓ | **30.1** | **41.4** | 33.6 |
| Baseline | ✓ | 28.4 | 39.1 | 32.4 |
| IMTL | - | 29.3 | 40.6 | 33.4 |
| IMTL-KD | - | 29.7 | 41.1 | **33.9** |

Table 2: Performance on different data set with additional training data.

achieve this, we propose the L2G feature extractor. We aim to use the interaction window size to limit the positions from which information is extracted. Convolution networks are well-suited for this purpose, and we implement the L2G extractor using:

$$\mathbf{x} = \mathbf{x} + \text{Conv}(\text{Norm}(\mathbf{x})) \qquad (7)$$

where $\text{Conv}()$ denotes the depthwise separable convolution (Chollet, 2016). We add the extractor in front of each Transformer layer in T-Enc. This extractor can learn relevant information from a given window $c$, which is determined by the convolution kernel size. Unlike the self-attention mechanism that learns from the entire sequence, this window focuses on a specific region, aiding the two tasks in learning the same information. However, it also introduces additional information for MT task, which necessitates the text's ability to enhance its denoising capabilities. Finally, we utilize the consistency loss to align the representations extracted by the extractor and attention mechanisms.

The study conducted by Xu et al. (2021) demonstrates that the MT task requires a more global understanding to form a semantic-level representation, whereas the acoustic task primarily relies on

local information. To address this, we propose an increasing window approach to assist the acoustic representation in capturing global textual information. Specifically, we introduce an increasing stride for the convolution field, where each layer's field increases by $d$. Therefore, the kernel size of the $i$-th T-Enc layer is $c + d * i$.

### 3.3 MTL Based on Task Impact

Our previous analysis reveals that the impact of different tasks and modules varies over time. This insight has inspired us to develop a new training strategy that gradually eliminates the auxiliary task, rather than relying on an additional fine-tuning stage. This approach simplifies and streamlines the entire training process. To achieve this objective, we need to determine whether the auxiliary task is beneficial at each training step and assess its level of impact. We can examine the change in task consistency to address the first question. When the task consistency stabilizes and different tasks reach a balanced state, we can reduce the training weight assigned to the auxiliary task. However, to effectively decrease the weight, we must quantify the influence of the auxiliary task.

In multi-task learning, the use of norms has been extensively studied (Argyriou et al., 2008; Maurer et al., 2013). Norms can evaluate the sparsity of a matrix and are commonly employed to enhance the information in network parameters, thereby improving the effectiveness of MTL. Consequently, gradient norms have been successfully utilized in computer vision (Chen et al., 2018) to balance the impact of different tasks. Taking inspiration from this, we propose a task impact metric for auxiliary tasks based on gradient norms. We sample $k$ instances from the training set to create $\mathcal{D}'$, which we then feed into the model to obtain gradients for the

| Models | En-De | Length ratio(%) | En-Fr | Length ratio(%) |
|--------|-------|-----------------|-------|-----------------|
| Baseline | 25.8 | 100.00 | 36.7 | 100.00 |
| Shrinking | 25.7 | 53.97 | 36.8 | 57.72 |
| +LBM | 26.3 | 55.67 | 37.2 | 60.13 |

Table 3: Ablation study on shrinking method.

| Models | En-De | En-Fr |
|--------|-------|-------|
| Baseline | 25.8 | 36.7 |
| +Fixed window | 26.2 | 37.3 |
| +L2G | 26.4 | 37.5 |
| LBM | 26.3 | 37.2 |
| +Fixed window | 26.6 | 37.5 |
| +L2G | 26.9 | 37.7 |

Table 4: Ablation study on L2G training.

various tasks. The task impact $m$ of auxiliary task $i$ can be calculated using the following formula:

$$m_i = \frac{1}{k} \sum_{j \in \mathcal{D}'} \left( \frac{||\delta_i^j||_2}{||\delta_{\text{st}}^j + \delta_i^j||_2} \right) \qquad (8)$$

where $\delta_i^j$ is the ATTEN (self-attention sub-layer) gradient of data $j$ for task $i$, $|| \cdot ||_2$ denotes the 2-norm of the matrix. The higher $m$ shows updating the gradient will have a greater impact on the ST task. Containing the change of different tasks, we give the weight of the different task at $t$-th update as follows:

$$w_i^t = w_i^{t-1}(m_i)^{u/s} \qquad (9)$$

where $u$ represents the current training step, and $s$ denotes the smoothing coefficient. The impact of these two hyper-parameters can be likened to temperature coefficients and we can set appropriate $u$ and $s$ values to ensure that changes in task weights correspond to changes in task consistency. Since the weight between T-Enc and Decoder differs, we select the maximum value as $w$ for the MT task. The design of $w$ takes into account the consistency and impact of different tasks, thus avoiding unnecessary computational resources when auxiliary tasks are not beneficial. Furthermore, this training strategy allows us to remove the other task in time and achieve optimal performance without the need for tedious fine-tuning stages.

## 4 Experiments

### 4.1 Data

We conducted experiments on the multilingual MuST-C dataset (Di Gangi et al., 2019). The dataset consists of eight language pairs: English (En) to German (De), French (Fr), Spanish (Es), Romanian (Ro), Russian (Ru), Italian (It), Portuguese (Pt), and Dutch (Nl). For the En-De, En-Fr, and En-Es MT tasks, we collected external training data from WMT16, WMT14, and WMT13 respectively. As additional ASR data, we utilized the LibriSpeech (Panayotov et al., 2015) clean-100

dataset. The Dev set was used for validation, and tst-COMMON set served as the test set for all tasks. SentencePiece[2] segmentation with a vocabulary size of 10,000 was applied to all training datasets. The detail of the data is shown in Appendix A.

### 4.2 Model settings

We used the Fairseq toolkit (Ott et al., 2019; Wang et al., 2020a) to implement our methods. The `Transformer-BASE` configurations were chosen as the baseline settings, with approximately 150M parameters. We reproduced the ConST method to establish a strong baseline (Ye et al., 2022). The acoustic encoder was initialized with the audio-only pre-trained HuBert (Hsu et al., 2021). In the presence of additional data, we followed the setup of SpeechUT (Zhang et al., 2022b), which utilized a hidden size of 768, 12 attention heads, and a 3072 FFN dimension. Each training batch contained 20M audio frames. We set the training steps to 80K. When using additional MT data, the data size for different tasks becomes extremely unbalanced. Therefore, we first trained the MT task for 15 epochs with 8192 tokens per batch and then sampled 3 million sentences as MT data for MTL. We change the updated frequency to 4 and the training step to 40K. The kernel size $c$ and the increased stride $d$ for the L2G extractor was set to 5 and 3, respectively. The value of $s$ was set to 5000 for the ASR task and 10,000 for the MT task. The initial weights of ASR and MT tasks are 1.0. We updated the task weight every 5000 training steps and removed the task when the weight fell below 0.1. During inference, we average the last 10 checkpoints for evaluation. The other decoding settings are the same as those in CRESS (Fang and Feng, 2023). We use ScareBLEU (Post, 2018) as the metric for ST performance. The experiments were conducted on eight NVIDIA GeForce RTX 3090 GPUs.

---

[2]https://github.com/google/sentencepiece

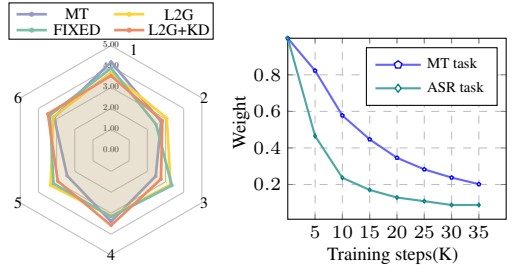

Figure 10: IE of different methods at the T-Enc layers (left). Task weights along training steps (right).

| Models | Training time | | |
|---|---|---|---|
| | En-De | En-Fr | En-Es |
| SpeechUT | 96 Gd + 80k tuning steps | | |
| IMTL | 12 Gd | 32 Gd | 20Gd |

Table 5: A comparison of training cost with additional MT data. 1 Gd indicates that using one GPU training one day. The SpeechUT and IMTL use the V100 and 3090 GPU respectively.

## 4.3 Results

The comparison of our IMTL and other works under the circumstance of no additional data is shown in Table 1. We find that the work utilizing the pre-training and fine-tuning paradigm achieves a significant improvement compared to the vanilla training strategy. M³ST even designs a two-stage fine-tuning method. However, few works have explored the extent of improvement gained by pre-training (Le et al., 2023), which is a high-cost method. Our IMTL, which dynamically decreases the weight of the auxiliary task and does not rely on fine-tuning, still achieves state-of-the-art (SOTA) performance. This proves that our method fixes the consistency during multi-task learning and further improves training efficiency. We have noticed that the newly proposed SOTA work implements teacher-forcing to bridge the modal gap, known as the knowledge distillation (KD) method. We further incorporate the KD method (Liu et al., 2019; Xu et al., 2021) into our IMTL, resulting in IMTL-KD. This demonstrates that our method is complementary to the KD method and achieves a new SOTA performance.

We also compare our method with other works that utilize extra training data. The SOTA work SpeechUT aims to cover all speech-to-text tasks, thus it requires a significant amount of training resources (pre-training for 3 days with 32 GPUs) and a complicated training strategy. In contrast, our model achieves comparable or better performance with much fewer training resources (e.g., 1.5 days with 8 GPUs for the En-De task) and does not require fine-tuning. The building process is much simpler and more efficient.

## 4.4 Effect of LBM

We compare the effects of the shrinking (Liu et al., 2020; Dong et al., 2021a) and LBM methods in Table 3. Directly using the shrinking method does not benefit the model, although it significantly reduces the length of the sequence. However, after applying the LBM method, the model achieves a 0.5 BLEU improvement while maintaining a low length ratio. This phenomenon demonstrates that shrinking alone is not stable, and the loss of information can lead to performance degradation. We find the average length of En-De audio is about two times the length of En-Fr audio, thus the shrinking effect is better.

## 4.5 Effect of L2G

We conducted an ablation study on L2G training, and the results are presented in Table 4. It shows that adding noise and constraining the field of information interaction significantly improve the performance compared to the baseline. Furthermore, the method still performs well based on the LBM, which confirms the conclusion that compressed sequences can learn additional information. When we apply the local-to-global strategy, the performance gains further improvement, which demonstrates that increasing the field size is more suitable for the goal of modal transformation.

We also analyzed the changes in information entropy (IE) when applying different methods in Figure 10. We observed that the IE of the first MT layer is the highest since we add some noise to the embedding. Compared to the fixed method, the L2G method can learn more information in the middle layers of the model, indicating that a fixed size hinders the extraction of more global information. After employing the KD method, the IEs of all layers become more consistent with MT, except for the first noisy layer.

## 4.6 Change of Task Weight

We display the changes in task weights in Figure 10. The weight of the ASR task decreases rapidly, while the weight of the MT task gradually decreases, slowly eliminating its impact on the ST task. This also aligns with the observed pattern of

gradient consistency in our analysis.

We compare the training time in Table 5 and find that our method requires about 12.5% ∼ 33.3% of the training cost of SpeechUT on three MuST-C tasks. Additionally, our method does not require alignment with the fine-tuning stage on the ST task. This demonstrates the efficiency of our method.

## 5 Related Work

E2E ST has gained attention for its advantages over cascade systems in terms of reduced latency and error propagation (Bérard et al., 2016; Duong et al., 2016; Weiss et al., 2017; Xu et al., 2023a). However, two main challenges hinder the adoption of E2E ST: 1) limited ST training data and 2) difficulties in modeling the modality gap. To address these challenges, pre-training strategies have emerged, including audio-only self-learning (Baevski et al., 2020; Hsu et al., 2021), joint audio-transcription encoding (Ao et al., 2022; Zhang et al., 2022b; Chen et al., 2022), and combining MT and ASR data for pre-training (Wang et al., 2020c; Zheng et al., 2021). These approaches have shown significant improvements in ST performance.

Pre-training methods are also combined with multi-stage and multi-task strategies. The multi-stage method involves pre-training all modules with auxiliary tasks, followed by integration and fine-tuning for the ST task (Xu et al., 2021; Li et al., 2021; Zhang et al., 2023). On the other hand, multi-task training utilizes multiple training objectives within a single model, eventually fine-tuning with the ST loss (Wang et al., 2020b; Le et al., 2020; Vydana et al., 2021; Tang et al., 2021; Ye et al., 2021). While most SOTA methods employ the pre-training and fine-tuning paradigm, few studies have investigated the impact of other tasks on boosting the ST task, considering the time-consuming nature of pre-training. Tang et al. (2022) provided a simple analysis that showed gradient interference is not serious and the effectiveness of MTL. In this paper, we conduct a comprehensive experiment to explore the impact and time efficiency of other tasks.

Mitigating differences in representation and addressing variations in sequence lengths are two ways used to bridge the modality gap between text and speech. Some work proposes the use of adapters to reduce differences in pre-trained modules (Bahar et al., 2019; Li et al., 2021; Xu et al., 2021). Contrastive learning (Ye et al., 2022; Zhang et al., 2023) and knowledge distillation tech-niques are also employed to achieve this objective (Fang et al., 2022; Zhou et al., 2023; Fang and Feng, 2023). Furthermore, the mixing-up of two modal representations has been found to be effective (Cheng et al., 2022). The inclusion of blank tokens (Wang et al., 2020b; Zhang et al., 2023) can improve denoising capabilities. To address length inconsistencies, shrinking based on ASR prediction or cluster methods have been utilized (Dong et al., 2021a; Zhang et al., 2022b).

## 6 Conclusion

Most advanced ST methods heavily rely on multi-task learning, but few studies focus on the relationship between auxiliary tasks and the ST task itself. In this study, we design a gradient consistency metric to analyze the impact of other tasks on the ST task during the multi-task learning process. Based on our analysis, we propose improved methods that address three key aspects: length, representation, and training efficiency. Experimental results on the MuST-C dataset demonstrate that our approach achieves state-of-the-art performance and significantly improves training efficiency.

## Acknowledgement

This work was supported in part by the National Science Foundation of China (No.62276056), the National Key R&D Program of China, the China HTRD Center Project (No.2020AAA0107904), the Natural Science Foundation of Liaoning Province of China (2022-KF-16-01), the Yunnan Provincial Major Science and Technology Special Plan Projects (No.202103AA080015), the Fundamental Research Funds for the Central Universities (Nos. N2216016, N2216001, and N2216002), and the Program of Introducing Talents of Discipline to Universities, Plan 111 (No.B16009). The authors would like to thank anonymous reviewers for their insightful comments.

## Limitations

There are some limitations that our work has not figured out. The analysis is mainly carried out on the MuST-C dataset, where the training data size is not large. We did not apply the state-of-the-art knowledge distillation (KD) method to further improve performance. The effect of knowledge distillation based on IMTL has not been sufficiently investigated.

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

# Appendix

## A  Data Details

We conducted experiments on the multilingual MuST-C dataset (Di Gangi et al., 2019). The detail of the data is shown in Table 6. The detail of additional data is shown in Table 7.

| Language | Hours(h) | Sentence(K) |
|---|---|---|
| En-De | 408 | 234 |
| En-Es | 504 | 270 |
| En-Fr | 492 | 280 |
| En-It | 465 | 258 |
| En-Nl | 442 | 253 |
| En-Pt | 385 | 211 |
| En-Ro | 432 | 240 |
| En-Ru | 489 | 207 |

Table 6: Training data size of the MuST-C 8 languages.

| Dateset | Language | Sentence |
|---|---|---|
| WMT16 | En-De | 3.9M |
| WMT13 | En-Es | 14.2M |
| WMT14 | En-Fr | 31.2M |
| LibriSpeeh 100h | En | 28.5K |

Table 7: Training data size of additional MT and ASR data.

## B  Contrastive Loss

In this paragraph, we introduce the notation and define the loss function for contrastive training. We start by defining two outputs: $\mathcal{A}(\mathbf{s})$ represents the output of the ST encoder when given the speech input $\mathbf{s}$, and $\mathcal{M}(\mathbf{x})$ represents the output of the pretrained text encoder when given the transcription $\mathbf{x}$. We then consider a set of training samples denoted as $(s_i, x_i)$.

The loss function for contrastive training, denoted as $\mathcal{L}_{\mathrm{CL}}$, is defined as follows:

$$\mathcal{L}_{\mathrm{CL}} = - \sum_{(s_i, x_i)} \log \frac{e^{\pi(\mathcal{A}(s_i), \mathcal{M}(x_i))/\tau}}{\sum_{x_j: j \neq i} e^{\pi(\mathcal{A}(s_i), \mathcal{M}(x_j))/\tau}}$$

(10)

In this equation, $\pi(\cdot, \cdot)$ is a function that computes the similarity between the input vectors. For our purposes, we choose the cosine function as

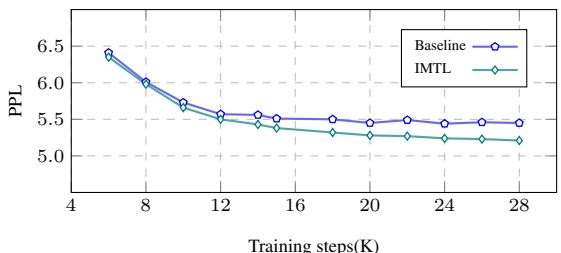

Figure 11: PPL of Dev set during training.

| Training task(s) | Speed (Seconds/Epoch) |
| --- | --- |
| ST, MT, ASR | ~1187 |
| ST, MT | ~936 |
| ST | ~675 |

Table 8: Training data size of additional MT and ASR data.

$\pi(\cdot, \cdot)$ and apply average pooling to the two sequence representations. The variable $\tau$ is a scaler that controls the sharpness of the function output, and in this case, we set $\tau$ to 0.1.

For each speech input $s_i$, we have its corresponding labeled transcription $x_i$, which forms a positive sample $(s_i, x_i)$. Additionally, we utilize transcriptions other than $x_i$ (denoted as $x_j$ for $j \neq i$) to create negative samples.

## C Information Entropy

Information entropy is a concept from information theory that measures the average amount of information contained in a set of data or the uncertainty associated with the data. In the context of information theory, entropy is calculated using the probabilities of different outcomes or events occurring within a system. The higher the entropy, the greater the uncertainty or lack of information about the outcomes. Conversely, lower entropy indicates a higher degree of predictability or knowledge about the outcomes. The formula is given by:

$$H(X) = -\sum p(x) * \log_2(p(x)) \quad (11)$$

where $H(X)$ represents the entropy of a random variable $X$, $P(x)$ is the probability of each possible outcome $x$, and the sum is taken over all possible outcomes.

## D Coverage Speed

Figure 11 shows the coverage speeds of the baseline and our IMTL. We can find the IMTL is better in terms of convergence speed and effect.

## E Training Speed

There are mainly three tasks (ASR, MT, and ST) during the training strategy. Our Improved Multi-Task Learning (IMTL) algorithm dynamically adjusts the training weights assigned to the auxiliary ASR and MT tasks. Specifically, any auxiliary task whose training weight diminishes below a threshold of 0.1 will be effectively halted to optimize the training process. As a bonus, subsequent training phases are computationally more efficient than the standard approach, given that both the forward and backward computations are integrated components of the overall training pipeline. Table 8 shows a rough estimate of the training speed of our IMTL approach on the MuST-C dataset with different training tasks.