# OpenReview forum: "Rethinking and Improving Multi-task Learning for End-to-end Speech Translation"
_EMNLP/2023/Conference — EMNLP 2023 Main_

### Official Review · Reviewer_idg1 · 2023-08-05

**Soundness:** 3

**Excitement:**

3: Ambivalent: It has merits (e.g., it reports state-of-the-art results, the idea is nice), but there are key weaknesses (e.g., it describes incremental work), and it can significantly benefit from another round of revision. However, I won't object to accepting it if my co-reviewers champion it.

**Paper Topic And Main Contributions:**

This paper investigates methods to enhance multi-task learning for end-to-end speech translation. The authors propose using cosine similarity between the gradient vectors computed from different tasks as a measure of consistency between tasks and present the following findings:

a) The ASR and ST tasks show high consistency with respect to the gradients at the acoustic encoder, but exhibit little consistency at the text encoder and decoder. Consequently, the authors suggest employing only the ASR task for training the acoustic encoder.

b) The consistency level between MT and ST is relatively low in both the text encoder and decoder, with the discrepancy being more pronounced at the text encoder. The authors hypothesize that the length discrepancy between audio features and text token sequences, along with the representation gap between the two, are sources of concern. They propose two approaches: one is a modified version of shrinking, aimed not to lose information, and the other involves adding speech-like noise (blanks and repetitions) to the text sequence, along with a local-to-global convolution approach to progressively utilize larger context windows.

c) The consistency between ST and the auxiliary tasks changes over the training process. The authors propose a task weighting strategy to adjust the importance of auxiliary tasks based on their consistency with the ST task, as well as the impact of the training gradient, removing the necessity of a separate fine-tuning step.

Experiments conducted on MuST-C demonstrate that the proposed approach achieves a 0.9 BLEU improvement over the ConST baseline, averaged over the 8 language pairs in MuST-C.

**Questions For The Authors:**

I recommend providing more descriptive explanations for the Figures so that they are self-contained and easier to read.

**Reasons To Accept:**

- The approach to investigate consistency between different tasks provides insights into the varied utilities of auxiliary tasks and helps identify potential areas of interest.
- Their proposed approaches, including the improved shrinking method, the local-to-global training method, and consistency/impact-based weighting, appear to be reasonable extensions and have been found to improve performance.
- The overall performance on MuST-C is decent.

**Reasons To Reject:**

- The writing needs improvement to provide a clear description of their proposed approaches. In its current form, it is not evident how their approach works and how to replicate the results. For example, how does the L2G extractor integrate with the Transformer and what are the configurations? What are "u" and "s" in equation 9?
- While the investigation of consistency motivates most of the proposed improvements, it is not clear how these methods affect or improve consistency, as no results were presented. Partially due to the writing, it is also not apparent why these proposed methods can address the identified problems.
- Some of the claims are not fully supported by their approach, partially due to writing. For instance, the authors suggest that it is necessary to consider both consistency and task impact in determining the weighting for the auxiliary task. However, their approach (equations 8 and 9) seems to only take into account the task impact, leaving task consistency unaddressed. Additionally, the authors claim that their approach can avoid unnecessary computation resources when the auxiliary tasks are not beneficial; however, their weight approach appears only to adjust the weights of different auxiliary tasks and does not reduce computation on these tasks.

**Reproducibility:**

3: Could reproduce the results with some difficulty. The settings of parameters are underspecified or subjectively determined; the training/evaluation data are not widely available.

**Reviewer Confidence:**

3: Pretty sure, but there's a chance I missed something. Although I have a good feel for this area in general, I did not carefully check the paper's details, e.g., the math, experimental design, or novelty.

---

> ### Author Rebuttal · Authors · 2023-08-28
>
> Thanks for your thorough comments sincerely. Overall, the reviewer stated that our paper is not self-contained enough due to the writing. This is due to page limitations, which prompted us to relocate the experiment details to the Appendix. We believe that most of your concerns can be addressed by referring to the Appendix directly. We will carefully revise the paper and add the essential details into experiments settings. Reviewer #3 commented that the paper is generally well-written and we suppose he has referred to the Appendix. Furthermore, we have submitted the codes for review and plan to open-source them. This will enable access to all relevant details and ensure the reproducibility of our results.
>
> ### The followings are response to Reasons To Reject:
> 1. **How do the authors introduce the L2G extractor and what is its configuration? What are "u" and "s" in equation 9?**
>
>     - Equation (7) details our L2G method, which serves as a key feature of our textual encoder. In an effort to enhance the encoder's capability for processing local information, we employ depthwise separable convolutions integrated within a residual network fashion. This design allows the model to transition seamlessly from capturing local patterns to understanding global context. The unique aspect of our L2G approach lies in the gradual expansion of the convolutional kernel size. This strategy facilitates the model's transition from focusing on local features to incorporating a more global field of view, thereby enriching the encoder's representational power. For a detailed configuration of our L2G method, including kernel sizes and other hyperparameters, we refer the reader to Appendix A.2, Line 872. The kernel size c and the increased stride d for the L2G extractor was set to 5 and 3 respectively.
>
>         Moreover, we have submitted our code as Supplementary Materials. The architecture of our model can be found at IMTL/fairseq/models/speech_to_text/s2t_joint.py, and the implementation of LBM is available here as the LookBackModule class on line 736. The implementation of L2G is located in IMTL/fairseq/models/transformer/transformer_encoder.py, spanning from line 100 to line 119.
>
>    - The coefficients 'u' and 's' in Equation (9) serve as parameters to regulate task weights. We emphasize our consideration of both task impact and consistency. Consequently, we select appropriate 'u' and 's' values to ensure that changes in task weights correspond to changes in task consistency. The impact of these two hyperparameters can be likened to temperature coefficients. Higher values of 'u' and lower values of 's' signify a faster decrease in task weight. Detailed configurations for 'u' and 's' are provided in Appendices L874 and L877, respectively. Specifically, 's' was set to 5000 for the ASR task and 10,000 for the MT task. 'u' was updated every 5000 steps to adjust the weights.
>
> 2. **The effect of proposed method on the consistency is not shown and the authors do not explain why their method could work.**
>
>     We appreciate the opportunity to elucidate how our proposed methods affect consistency and why they yield effective results. Note that our proposed method is inspired by our observation and the subsequent analyses, thus some changes in consistency have shown in the analysis section, such as the impact of removing the ASR task from the decoder, as depicted in Figure 4. Regarding why our method effectively resolves the identified problem, we have indeed provided explanations:
>
>    - **Impact of LBM on Consistency**: As shown in Table 3, the introduction of the LBM enhances the frame length by approximately 3%, leading to an improvement in performance. Specifically addressing consistency, Figure 5, labeled as "Length," illustrates that the shrinking method displays better consistency in the decoder.
>
>    - **Role of L2G in Task Representation**: The Local-to-Global (L2G) method aims to bridge the gap in task representations between ST and MT. While the impact on consistency was initially elusive, we applied an information entropy metric for a more nuanced evaluation. Figure 10 (left) reveals that the L2G and L2G-KD methods result in less disparity in task representations compared to traditional fixed window methods.
>
>    - **Impact of Task Weight Reduction**: We have provided the training perplexity (PPL) as a measure of model performance after applying task weight adjustments. These findings could be found in Appendix D, confirm that our method achieves a faster convergence rate.
>
>     We believe the aforementioned explanations adequately elucidate the efficacy of our method, and we will supply changes in consistency after employing the complete method in the subsequent version.
>
> 3. Some of the claims are not fully supported by their approach.
>
>     - **The approach (equations 8 and 9) seems to leave task consistency unaddressed**: Equation 9 takes consistency into account, and in L425, 'u' and 's' represent the coefficients which are formulated based on the consistency alteration. Additionally, we can adjust the 'u' and 's' values to control the rate of task weight reduction. For instance, Figure 7 illustrates the rapid decrease in consistency for the ASR task. In response, we will enhance the update frequency accordingly. Consequently, the weight of the ASR task adheres to the evolving consistency patterns of the ASR task.
>
>     - **Adjusting the weights of different auxiliary tasks can not reduce computation**: Regarding the reason for the efficacy of weight adjustment in expediting the training process, we specify in L879 that we cease training a task if its weight drops below 0.1. This approach prevents needless training. As illustrated in Figure 10 (right), we halt training the ASR task after 35,000 steps. Further, Figure 11 in Appendix D shows a faster convergence speed.
>
>     We suppose that these issues might have arisen due to moving the experiment settings to the Appendix section. To address this, we intend to provide additional descriptions in the main content to ensure our claims are self-contained.
>
> ### Response to Questions:
>
> We will add sufficient explanations and make the Figures easy to understand for readers. If you have any further questions about our work, please feel free to contact us during the discussion period.

---

### Official Review · Reviewer_uUBZ · 2023-08-10

**Soundness:** 4

**Excitement:**

4: Strong: This paper deepens the understanding of some phenomenon or lowers the barriers to an existing research direction.

**Missing References:**

In addition to the important references cited in "Reasons to reject", I think the literature review could be improved by the following additional references:
- For shrinking method, should cite the first work (as far as I know) proposing this, which is Liu et al. 2020 "Consecutive decoding for speech-to-text translation".
- L208: Previous work on ST earlier than Fang et al., 2022 already highlighted this and proposed different methods to remedy this issue such as Li et al., 2020 ("Multilingual speech translation with efficient finetuning of pre-trained models").
- Landmark papers on multi-task learning such as Bahar et al. 2019 (A Comparative Study on End-To-End Speech to Text Translation) who used CTC loss for ASR task in joint training should be appreciated.
- L446: Le et al., 2023 (Pre-training for Speech Translation: CTC Meets Optimal Transport) studied the effects of different losses in supervised pre-training.
- L560:  Bahar et al. 2019 (A Comparative Study on End-To-End Speech to Text Translation), Li et al., 2020 ("Multilingual speech translation with efficient finetuning of pre-trained models") proposed using adapters to connect the pre-trained modules while Le et al., 2021 studied the use of adapters within every Transformer layers.

**Paper Topic And Main Contributions:**

This paper performs a comprehensive study of the effects of auxiliary tasks (namely ASR and MT) to the ST task in a multi-task learning (MTL) setup. Based on the observations from the analyses, the authors propose an MTL approach that specifically improves the consistency between the tasks, mitigates the length mismatch, and bridge the representation gap due to different modalities. The proposed method consists of three main parts:
- A shrinking-based method called Looking-back mechanism (LBM) based on CTC outputs to minimize information loss
- Local-to-global Training (L2G) strategy that inserts audio-like noises into the text input sequence, following by a CNN extractor to enable the consistency of the representation learning between the two tasks
- MTL Based on Task Impact training strategy that adjusts the task impacts as training progresses.

**Questions For The Authors:**

- Please refer to the first point in "Reasons to reject". Do the reported result uses any form of (both supervised and self-supervised) pre-training? If it does use some form of pre-training, why do you specify "no fine-tuning" (Table 1) to characterize the proposed method?
- L119: Do you mean that the ASR encoder is shared with ST encoder, while the three decoders of the ASR, MT, and ST tasks are shared?
- There is no embedding layer for text in Figure 9. Is the text input directly fed to the Transformer decoder? This is a bit unusual in my opinion. Could you explain this choice of configuration?

**Reasons To Accept:**

- The paper is well-written in general. The structure is clear and easy to understand.
- The work is well-motivated as MTL is currently a dominant approach in building strong ST systems, hence a good understanding of the interplay between the auxiliary tasks with the main ST task would facilitate researches in this direction.
- While many of the findings/observations from this paper have been known in previous work, the paper offers an interesting quantitative analysis based on module gradients and information entropy to systematically demonstrate the benefits and/or downsides of each auxiliary task (ASR/MT) to ST.

**Reasons To Reject:**

1. The scope of work is confusing in several parts and some key details are missing.
	- As I understand, L57-62 discusses MTL in (both self-supervised and supervised) pre-training, but the paper focuses on MTL in ST training which usually requires no additional fine-tuning. Does it mean that no pre-training in any form is used in this work?
	- This paper examines ASR's impact on the speech encoder (L67, 195) within the ST joint training framework. Yet, the references to Tang et al. 2022 and Zhang et al. 2022c (L195) pertain to pre-training.
	- The model architecture used as well as other important details on experiments are not included in the main content for easier comparison with previous work.

2. Previous work that the proposed method built upon is not clearly specified in the method description.
	- Previous CTC-based shrinking method, which is referred to as unstable shrinking in the paper and is the first step in the proposed LBM method, should be mentioned in related text to enhance clarity.
	- Likewise, proper credit to prior research like Wang et al. 2020b for the idea of introducing noise to text input should be included in the relevant section (description of the L2G approach).

3. Several findings are a bit weakly positioned with existing work; additional references are needed to enhance transparency. In particular, it is not clear from the Introduction (and in later text) that which findings have already been known and the differences in contributions this paper makes compared to previous work should be emphasized.
	- Regarding "ASR aids the acoustic encoder", this is a well-established fact and previous works that proved this result in a MTL setup (which is the main topic of this paper) should be appreciated such as Weiss et al. 2017, Anastasopoulos et al. 2018 (Tied Multitask Learning for Neural Speech Translation), and Bahar et al. 2019 (A Comparative Study on End-To-End Speech to Text Translation).
	- Regarding "MT facilitates the textual encoder in audio-to-text transfer", this is the motivation behind the increasingly common cascaded encoders approach to decouple the speech encoder into cascaded speech-then-text encoders. Initial work in this direction should be mentioned such as Liu et al. 2020 (Bridging the Modality Gap for Speech-to-Text Translation), Dong et al. 2021 (Listen, Understand and Translate": Triple Supervision Decouples End-to-end Speech-to-text Translation), and Xu et al. 2021.
	- "length inconsistency hinders aligning the representations of the two modalities". The length inconsistency issue has been raised in previous work such as Wang et al. 2020b  and Liu et al. 2020 (Bridging the Modality Gap for Speech-to-Text Translation).

**Reproducibility:**

3: Could reproduce the results with some difficulty. The settings of parameters are underspecified or subjectively determined; the training/evaluation data are not widely available.

**Reviewer Confidence:**

4: Quite sure. I tried to check the important points carefully. It's unlikely, though conceivable, that I missed something that should affect my ratings.

**Typos Grammar Style And Presentation Improvements:**

- It would facilitate the reading to briefly explain "task consistency" in the Introduction in my opinion.
- L316: $w$ is not previously defined. Similar for ATTN in L418.
- As a minor suggestion, previous work such as Tang et al. 2022 that also adopts the same analysis method using gradient similarity between subtasks should be cited in the relevant part as well.
- Figure 5-6, what do legends such as "Baseline", "Length", and "Rep" mean?
- L173: typo in the full stop in the middle of a sentence.
- L246: "needs mitigate" -> "needs to be mitigated"
- L279: "two main problems stall need improve" -> "two main problems still need to be improved"?
- L283: "guide the acoustic encode" -> "guide the acoustic encoder"
- L292: link to Figure 3 needs to be fixed (it leads to Table 3 instead).
- L336: "a information difference" -> "an information difference"
- Eq 8: as a minor suggestion, putting 1/k outside of the sum is more visual in my opinion.
- Table 2: "STFT" -> "STPT"

---

> ### Author Rebuttal · Authors · 2023-08-28
>
> Thanks sincerely for your thorough and valuable feedback for further improving our paper. We think that we can address them all in the improved version of the paper.
>
> First and foremost, the reviewer expressed concern about why we still report certain well-known findings. Therefore, we reiterate our motivation here. The field of end-to-end ST has been progressing rapidly, with the application of self-supervised pre-trained models gaining traction. State-of-the-art (SOTA) performance is also advancing swiftly, leading to more intricate training strategies and higher training costs simultaneously. Given this context, we aim to reconsider whether the prevailing multi-task learning approach still aligns with previous conclusions. Furthermore, we aim to propose solutions that enhance the training strategy.
>
> For instance, what is the optimal way to employ the ASR task? While certain prior studies have demonstrated the usefulness of the ASR task in the encoder, recent SOTA research has yielded contrasting outcomes. For instance, progressive training methodologies such as XSTNet (Ye et al., 2021) and ConST (Ye et al., 2022) relocate the ASR task to the decoder, achieving state-of-the-art (SOTA) performance. Conversely, some studies have even omitted the ASR task altogether, with methodologies like STEMM (Fang et al., 2022), CMOT (Zhou et al., 2023), and CRESS (Fang et al., 2023) still managing to secure SOTA results. These conflicting results raise intriguing questions, motivating us to undertake a comprehensive analysis to clarify the role and optimal positioning of the ASR task in modern multimodal systems.
>
> Secondly, we must acknowledge that certain experimental settings are not included in the main content due to page limitations. However, we have presented our training configuration in the Appendix and have included the training details in the code submitted alongside this paper.
>
> The followings are our response to all you concerned.
>
> ### Response to Reasons To Reject:
> 1. Some details are not clear enough.
>     - **Does it mean that no pre-training in any form is used in this work?**
>
>         In our experiments, as specified in line 859, we employ Hubert, a self-supervised speech feature extractor, for the acoustic encoding. It's noteworthy that we do not consider the use of this pre-trained model as part of our multi-task learning (MTL) framework. This follows the convention in existing literature where such self-supervised models are treated solely as feature extractors and are not counted in the pre-training phase.
>
>        We wish to clarify that our decision to employ Hubert arises from the challenges posed by the substantial noise in raw audio data. Preliminary attempts to train an end-to-end Speech-to-Text (ST) model from scratch were not successful. Nonetheless, we recognize the potential of using Mel filterbanks as an alternative speech feature. We remain committed to exploring this avenue in future work.
>
>    - **The two cited papers are not appropriate enough, because they used pre-training while here is multi-task learning.**
>
>         We think that the paper [1] is more suitable to be referenced in this context. Their work demonstrates the utility of incorporating CTC during the pre-training phase. Additionally, in our perspective, STPT (Tang et al., 2022), and SpeechUT (Zhang et al., 2022c) have also employed multi-task learning during their pre-training stages. Concretly, STPT (Tang et al., 2022) utilizes Speech to Phoneme classification task and SpeechUT (Zhang et al.,2022c) uses Speech-to-Unit task the as the ASR task respectively. We cite these studies to substantiate that ASR remains beneficial, particularly within the context of multi-task learning, as some work has removed the ASR task from their strategy.
>
>      [1] Pre-training for Speech Translation: CTC Meets Optimal Transport. Le et al., 2023. ICML 2023
>
>     - **Some important details on experiments are not included in the main content.**
>
>          Due to page limitations, we were unable to include experimental details in the main content. However, as described in the Appendix, our baseline is identical to that of ConST, and the training configuration and data used in conjunction with external MT data match those of SpeechUT, ensuring a fair comparison. Our code will be made available as open-source, and comprehensive training details can be found in the Supplementary Materials submitted alongside this paper.
>
> 2. **Certain previous work is missing when introducing the proposed method.**
>
>     We will revise the sections that are not adequately cited. Thanks for these suggestions to make our paper clear.
>
> 3. **Several findings are not novel enough compared with existing work.**
>    - **Regarding the finding that "ASR aids the acoustic encoder"**, we have explained the rationale at the beginning of this response. Some studies have omitted the ASR task, prompting us to explore the most effective utilization of the ASR task.
>    - **Regarding the finding that "MT facilitates the textual encoder in audio-to-text transfer"**, we perceive this discovery as an explanation for the phenomenon depicted in Figure 3 and as the basis for the subsequent analysis.
>    - **Regarding the finding that "length inconsistency hinders aligning the representations of the two modalities"**, this paper seeks to underscore the significance of length inconsistency in influencing the alignment of representations between the two modalities. This observation is often overlooked in works aiming to bridge the gap between the two modalities.
>
> ### Response to Questions:
> - **Do you use any form of (both supervised and self-supervised) pre-training?**
>
>     As stated in Appendix A.2, we utilize the pre-trained HuBert model. Many works consider it as a feature extractor and do not incorporate it into the pre-training stage. The reason we emphasize that our approach does not necessitate a fine-tuning stage is that we have observed several other methods requiring additional pre-training strategies (e.g., SpeechUT, Progressive training), followed by fine-tuning, apart from using a pre-trained acoustic encoder. These approaches are complex and time-consuming. Therefore, we introduce an improved MTL approach that yields the ST model directly, eliminating the need for intricate pre-training processes. Furthermore, previous studies also overlook accounting for the pre-training time of HuBert, making the comparison of training costs with their methods equitable.
> - **Do you mean that the ASR encoder is shared with ST encoder, while the three decoders of the ASR, MT, and ST tasks are shared?**
>
>     Regarding the baseline architecture, the answer is yes. The ASR task shares all three modules—acoustic encoder, textual encoder, and decoder—with the ST task. Similarly, MT also shares two modules—textual encoder and decoder—with the ST task. The baseline architecture is the same as ConST (Ye et al., 2022).
>     In contrast, for our IMTL approach, the ASR task only comprises a single acoustic encoder that is shared with the ST task. We arrived at this configuration based on our analysis, as we determined that the ASR component is unnecessary for the decoder.
>
> - **The description of Figure 9 is not clear.**
>
>     Figure 9 depicts the architecture of the encoder components. The left part represents the two encoders for the ST task, while the right part corresponds to the textual encoder for the MT task, with the source text serving as input. The textual encoder is shared between the two tasks, and both of their outputs are subsequently fed into the decoder (which is shared too and not illustrated in this figure) through cross-attention modules. Additional description will be incorporated to clarify these details.
>
> ### Response to Typos Grammar Style And Presentation Improvements:
> **What do the legends, such as "Baseline," "Length," and "Rep," mean in Figures 5 and 6?**
>
> "Length" refers to the model utilizing the shrinking method (L209). On the other hand, "Rep" pertains to the model incorporating the contrastive learning (CL) loss positioned at the top of the T-Enc method (L211). The CL loss serves as an alignment mechanism operating on the representation (Rep).

---

### Official Review · Reviewer_SFpv · 2023-08-11

**Soundness:** 4

**Excitement:**

3: Ambivalent: It has merits (e.g., it reports state-of-the-art results, the idea is nice), but there are key weaknesses (e.g., it describes incremental work), and it can significantly benefit from another round of revision. However, I won't object to accepting it if my co-reviewers champion it.

**Paper Topic And Main Contributions:**

In this paper, auhtors investigate the consistency between different tasks, considering different times and modules. They find that the textual encoder primarily facilitates cross-modal conversion, but the presence of noise in speech impedes the consistency between text and speech representations. Furthermore, auhtors propose an improved multi-task learning (IMTL) approach for the ST task, which bridges the modal gap by mitigating the difference in length and representation.

**Reasons To Accept:**

1. The proposed method attains state-of-the-art results.
2. The problem is well analyzed.
3. Experiments are abundant

**Reasons To Reject:**

1. The instability training of the ctc shrink mechanism can actually be solved by simple pre-training on the ASR task [1]. LBM doesn't seem necessary.
2. LBG focuses on a specific region. Limiting the visible context of attention can also solve this problem.

[1] Bridging the Modality Gap for Speech-to-Text Translation. arXiv:2010.14920v1

**Reproducibility:**

3: Could reproduce the results with some difficulty. The settings of parameters are underspecified or subjectively determined; the training/evaluation data are not widely available.

**Reviewer Confidence:**

4: Quite sure. I tried to check the important points carefully. It's unlikely, though conceivable, that I missed something that should affect my ratings.

---

> ### Author Rebuttal · Authors · 2023-08-28
>
> Thanks sincerely for your thorough comments, we think that we can address them all in the improved version of the paper. Here is our response to each of the concerns.
>
> ### Response to the weaknesses:
> 1. **The instability training of CTC shrink mechanism can be solved by pre-training the ASR task.**
>
>     We appreciate the reviewer's insightful comment on the training instability of the CTC Shrinking mechanism and its potential resolution through pre-training on the ASR task. We acknowledge the merit in this approach and would like to highlight that our "Shrinking" method is indeed congruent with the strategy presented by Liu et al. In the improved version, we will add this reference and make a more comprehensive comparison with our method. Here, we would like to highlight the differences between Liu et al. 2020’s work with ours:
>     - Our primary motivation is to simplify the often complicated training process. As a result, our novel LBM method successfully achieves training stability via multi-task learning, obviating the need for a pre-trained ASR model. We think this is an important contribution, in line with our aims to streamline the training process.
>     - Furthermore, we'd like to draw attention to the empirical results presented in Table 3, which underscore the superior performance of our LBM approach. Unlike the "Shrinking" method, which only attains a performance similar to the baseline, our LBM retains more essential information in the sequence, leading to a more robust performance, e.g. 26.3. In addition to resolving the issue of training instability, our LBM also effectively mitigates the loss of important sequence information, a problem that remains unaddressed by the "Shrinking" method alone.
>
> 2. **Limiting the visible context of attention can also solve this problem.**
>
>     Thank you for pointing out the alternative approach of limiting the visible context of attention as a potential solution. To provide a rigorous comparison, we implemented this strategy using the cross-attention module, as you suggested. Specifically, we modified the temperature parameter in the softmax function within the attention operation to narrow the focus of the distribution, thereby "limiting the visible context of attention."
>
>     We evaluated this approach on the MuST-C En-De dataset alongside our LBM and the baseline. Contrary to expectations, the method of limiting attention did not yield a significant improvement in performance. This led us to explore other avenues, culminating in the development of our LBM method. By integrating additional information into the shrunken sequence, our LBM method achieved much better performance, underscoring its simplicity and effectiveness. In light of your suggestion, we will include a more comprehensive comparison between the method of limiting attention and our LBM in the updated version of the paper.
>
>     | System | MuST-C En-De |
>     | ------ | --------------- |
>     | Baseline (Ye et al., 2022) |     25.8      |
>     | Shrinking (Liu et al., 2020)  |     25.7       |
>     | Limiting attention weight|25.9|
>     | LBM (Ours)                  |26.3|

---

### Official Review · Reviewer_XztX · 2023-08-11

**Typos Grammar Style And Presentation Improvements:** line 292
**Soundness:** 4

**Excitement:**

3: Ambivalent: It has merits (e.g., it reports state-of-the-art results, the idea is nice), but there are key weaknesses (e.g., it describes incremental work), and it can significantly benefit from another round of revision. However, I won't object to accepting it if my co-reviewers champion it.

**Paper Topic And Main Contributions:**

This paper conducts a comprehensively the impact of other tasks on the ST task during the multi-task learning process. This paper propose an improved multi-task learning (IMTL) approach for the ST task, which bridges the modal gap by mitigating the difference in length and representation.

**Reasons To Accept:**

1 I really like the deep analysis on Task Consistency.

2 Problem formulation is well defined and the results seems to perform better.

**Reasons To Reject:**

This paper lacks an organized model diagram to help readers better understand the entire algorithmic process

**Reproducibility:**

4: Could mostly reproduce the results, but there may be some variation because of sample variance or minor variations in their interpretation of the protocol or method.

**Reviewer Confidence:**

3: Pretty sure, but there's a chance I missed something. Although I have a good feel for this area in general, I did not carefully check the paper's details, e.g., the math, experimental design, or novelty.

---

> ### Author Rebuttal · Authors · 2023-08-28
>
> Thank you sincerely for your valuable comments, for which we are deeply grateful. Here is our response to them. Here is our response to them.
>
> ### Response to the weakness:
>
> We appreciate your suggestion regarding the incorporation of an organized model diagram to convey our approach more effectively. In the current version of our paper, Figure 1 offers a concise overview of a widely-used multi-task training architecture. This architecture comprises an acoustic encoder designed to process speech input, followed by a textual encoder that takes both the output from the acoustic encoder and the source text. The architecture culminates in a decoder that computes three task-specific losses: for Automatic Speech Recognition (ASR), Machine Translation (MT), and Speech Translation (ST), respectively.
>
> While Figure 1 serves as a foundational framework, we recognize the importance of a more holistic and clearer diagram to elucidate our specific contributions. Specifically, our Shrinking Mechanism and LBM module, as illustrated in Figure 8, are injected between the two encoders. LBM is also based on the computation of the auxiliary loss in the original architecture. Additionally, our Local-to-Global Training Strategy, presented in Figure 9, serves as an enhancement to the textual encoder originally depicted in Figure 1. In the revised version of the paper, we intend to integrate a more encompassing architecture overview that synthesizes all these elements for clearer, more intuitive understanding.

---

### Meta-Review · Area_Chair_r6o4 · 2023-09-19

**Recommendation:** 4

**Metareview:**

This paper studies the impact of auxiliary tasks (ASR, MT) on speech translation in the multi-task learning (MTL) setup. The authors presents in-depth analysis on the interaction between tasks and identifies issues causing degradation. The authors propose an improved MTL approach with a number of techniques (looking-back mechanism based on CTC output, local-to-global strategy to insert audio-like noises, dynamic adjustment based on task impact).

The paper is well positioned, and all reviewers enjoy the comprehensive and systematic analyses for deeper understanding of how each auxiliary task benefits or harms the primary task. The proposed improved MTL approach are also evaluated extensively and show favorable performance over many baselines. Authors have addressed most of the comments regarding experimental setups and positioning of the paper compared to the literature. I recommend the authors to incorporate the feedback from the reviewers into the manuscript.

---

### Decision · Program_Chairs · 2023-10-07

**Decision:**

Accept-Main

**Comment:**

This paper studies the impact of auxiliary tasks (ASR, MT) on speech translation in the multi-task learning (MTL) setup. The authors presents in-depth analysis on the interaction between tasks and identifies issues causing degradation. The authors propose an improved MTL approach with a number of techniques (looking-back mechanism based on CTC output, local-to-global strategy to insert audio-like noises, dynamic adjustment based on task impact).

The paper is well positioned, and all reviewers enjoy the comprehensive and systematic analyses for deeper understanding of how each auxiliary task benefits or harms the primary task. The proposed improved MTL approach are also evaluated extensively and show favorable performance over many baselines. Authors have addressed most of the comments regarding experimental setups and positioning of the paper compared to the literature. I recommend the authors to incorporate the feedback from the reviewers into the manuscript.